# Annual Direct Cost and Cost-Drivers of Systemic Lupus Erythematosus: A Multi-Center Cross-Sectional Study from CSTAR Registry

**DOI:** 10.3390/ijerph20043522

**Published:** 2023-02-16

**Authors:** Haiyan Wang, Mengtao Li, Kaiwen Zou, Yilin Wang, Qiaoling Jia, Li Wang, Jiuliang Zhao, Chanyuan Wu, Qian Wang, Xinping Tian, Yanhong Wang, Xiaofeng Zeng

**Affiliations:** 1Department of Epidemiology and Biostatistics, Institute of Basic Medical Sciences, Chinese Academy of Medical Sciences & School of Basic Medicine Peking Union Medical College, Beijing 100005, China; 2Department of Rheumatology and Clinical Immunology, Chinese Academy of Medical Sciences & Peking Union Medical College, National Clinical Research Center for Dermatologic and Immunologic Diseases (NCRC-DID), Ministry of Science & Technology, State Key Laboratory of Complex Severe and Rare Diseases, Peking Union Medical College Hospital (PUMCH), Key Laboratory of Rheumatology and Clinical Immunology, Ministry of Education, Beijing 100730, China

**Keywords:** systemic lupus erythematosus, direct costs, risk factor

## Abstract

Background: To estimate the annual direct costs and cost-drivers associated with systemic lupus erythematosus (SLE) patients in China. Methods: A multi-center, cross-sectional study was conducted based on the CSTAR registry. The information on demography and expenditures for outpatient and inpatient visits due to SLE were collected using online questionnaires. These patients’ medical records were from the database of the Chinese Rheumatology Information System (CRIS). The average direct costs and 95% confidence interval were estimated using the bootstrap method with 1000 bootstrap samples by resampling with replacement. The cost-drivers were identified using multivariate regression models. Results: A total of 1778 SLE patients from 101 hospitals participated in our study, with 92.58% as females, a mean age of 33.8 years old, a median duration of SLE of 4.9 years, 63.8% in an active disease state, 77.3% with two organs or more damaged, and 8.3% using biologics as treatment. The average annual direct cost per patient was estimated at CNY 29,727, which approximates to 86% for direct medical costs. For moderate to severe disease activities, the use of biologics, hospitalization, treatment of moderate or high dose glucocorticoids, and peripheral vascular, cardiovascular, and/or renal system involvements were found to substantially increase the direct costs, while health insurance slightly decreased the direct costs of SLE. Conclusions: This study provided reliable insight into financial pressures on individual SLE patients in China. The efforts focusing on preventing flare occurrences and limiting disease progression were recommended to further reduce the direct cost of SLE.

## 1. Introduction

Systemic lupus erythematosus (SLE) is a chronic complicated autoimmune disorder with multi-systemic involvements, characterized by recurrent flareups and subsequent remissions. Approximately 33–50% of patients will develop irreversible organ damage within 5 years of diagnosis [1,2,3]. SLE predominantly affects women of childbearing age and the peak age of onset is after their late teens and before their early forties [4,5,6]. Moreover, it is a currently incurable, but treatable disease. The patients have to take long-term medication in order to control disease activity, prevent disease flareups, and delay disease progression. Furthermore, the severe manifestations of SLE and the potential complications from therapy can lead to frequent hospitalizations. All of these might significantly increase health resource utilization and expenditures attributable to SLE.

The direct cost of SLE represents the costs of all kinds of resources used to treat a disease, including medical and non-medical costs [7]. The former refers to the therapeutic related inputs (including costs of the diagnosis, treatment, continuing care, emergency care, and rehabilitation, etc.), and the latter refers to costs associated with a disease, although not medical in nature (e.g., commuting and accommodation costs, etc.) [7]. The direct cost per patient serves as one of the basic elements to further estimate the monetary burden of disease imposed on society and to conduct economic evaluations of different treatment or management strategies. These, alongside the distribution of the components, also provide objective and quantitative evidence regarding the economic impact of the disease on patients or families, which are necessary for policymaking, designing, and managing health plans to protect patients from catastrophic costs.

Until now, the previously published evidence available for direct costs of SLE were mainly from only several countries or regions, including the US, Canada, the UK, Germany, Greece, Japan, Korea, Hong Kong, and Taiwan [8,9,10,11,12,13,14,15,16]. The overall average annual direct costs per patient, estimated by using patients’ self-reported data, varied dramatically ranging from $3735 to $14,410, from one country to another [17], which were due to the significant differences in the cost framework, methods of costs calculations, healthcare delivery systems, and patterns of practice [8,9,10,11,18,19]. Some cost-drivers (e.g., young age, high disease activity, more organs damaged, exacerbation of the disease, flares, impaired physical or mental health due to SLE, etc.), were also identified in previous studies; however, there have been disparities across studies [8,20,21,22,23]. The patients with SLE were different in the clinical features, socioeconomic status, ethnic profiles, coverage of health insurance, access to health resources, etc., which accounted for the disparities in these studies, to a certain extent.

However, accurate information on the direct cost of SLE and cost-drivers in China remains unavailable. A study in 2017 by Zhang, et al. reported the average annual direct costs of SLE were CNY 33,899 per patient among 121 outpatients in Shanghai, China [24]. Huang et al. reported in 2020 that the median of the direct costs was CNY 15,520 among 77 SLE inpatients from two hospitals in the Anhui province [25]. These findings hardly described the whole profile of costs in China because they were limited, including too small sample sizes, and were only representative of a few hospitals in the local areas.

Therefore, the aim of this study was to estimate the annual direct cost of SLE and cost-drivers in China through a multi-center, cross-sectional survey of patients from the Chinese Systemic Lupus Erythematosus Treatment and Research Group (CSTAR) Registry. This information is needed for policymakers and clinicians to determine the optimal allocation of healthcare resources, formulate public health programs, set priorities for disease prevention and control, and optimize treatment strategies, in order to achieve the ultimate goal of improving patients’ outcomes.

## 2. Materials and Methods

### 2.1. Study Design

This observational study was conducted from December 2020 to May 2022. The patients in our study were registered at the Chinese Rheumatology Data Center (CRDC), which was established in 2011 based on the Chinese Systemic Lupus Erythematosus Treatment and Research Group (CSTAR). We sent online survey invitations to patients who fulfilled the 1997 SLE classification criteria revised by the American Rheumatology Association (ACR), or the 2012 SLICC classification criteria for SLE, and had updated their clinical records on CRDC during the last 30 days. During the period of study, there were 18,163 patients who met the criteria of recruitment. Of them, a total of 1778 patients (about 9.79%), from 101 state-owned hospitals in 27 provinces of China, accepted the invitation and participated in this survey (Figure 1). After signing the informed consent form, they filled in the online questionnaire focusing on the expenditures of SLE during the year prior to the survey. The study was approved by the institutional review board of the institute of Basic Medical Sciences, Chinese Academy of Medical Sciences (Project No. 063-2020).

### 2.2. Data Collection

We used an online questionnaire to collect the cost data related to SLE. The questionnaire included patients’ demographic information and expenditures regarding outpatient and inpatient visits due to SLE, which had occurred one year prior to the survey. For each patient of SLE, the expenditures for the recent outpatient visits were collected, including medications, ancillary services (laboratory tests, radiology tests, and antibody tests), physician charges, transportation, and food expenses, as well as days lost from work related to medical visits or as an inpatient from SLE and for any accompanying personnel. The number of outpatient visits in the last year was also reported by the patients. For those who self-reported inpatient visits at least once in the last year, additional information about inpatient expenditures for visits was required to be recalled and collected (Figure 1). At the initial stage of the study, 53 participants were randomly selected to collect information repeatedly over a two-week period to assess the agreement of information based on the hypothesis that there was only a slight variation of costs recalled for the same visit.

Furthermore, we tracked these patients’ medical histories, the results of the laboratory tests, disease activity, organ damage assessments, and treatment regimens from the database of the Chinese Rheumatology Information System (CRIS) (Figure 1).

### 2.3. Measurements

The source of the patients was divided into three geographic areas (eastern, central, and western regions) according to the criteria of the National Bureau of Statistics of China. The disease duration was defined as the years from the onset of SLE diagnosis, which was classed into three groups (≤3 years, 3–8 years, and >8 years) by the quantile. Those patients whose disease duration was less than 3 months were identified as newly treated patients.

Disease activity and organ damage were evaluated using the SELENA-SLEDAI instrument and the SLICC/ACR index, respectively. The severity of the disease was divided into mild (SLEDAI score < 8) and moderate to severe (SLEDAI score ≥ 8). The patients with abnormal liver function were identified using metrics of alanine aminotransferase (ALT) > 40 U/L, or aspartate aminotransferase (AST) > 35 U/L, or alkaline phosphatase (ALP) > 130 U/L, or total bilirubin (TBiL) > 20 μmol/L. Low complement was defined as complement component 3 (C3) less than 0.73 g/L, or complement component 4 (C4) less than 0.1 g/L. Clinical remission was defined as a SLEDAI score of 0 and PhGA of <0.5, with an allowed glucocorticoid dose of ≤5 mg/day (prednisone or equivalent). Lupus low disease activity state (LLDAS) was defined as the following: (1) SLEDAI score of ≤4 with no scores for the renal, central nervous system, serositis, vasculitis, or constitutional components; (2) no increase in any component since the previous visit; (3) PhGA of ≤1; and (4) glucocorticoid dose of ≤7.5 mg/day (prednisone or equivalent).

### 2.4. Direct Cost Calculation

The costs of SLE were calculated in this study using a bottom-up approach (person-based data). For each patient, the total expenditures of SLE during the last year were the total outpatient and inpatient expenditures. The outpatient expenditures were calculated by the multiplication of recent outpatient visit costs and the number of visits during the last year estimated by patients (Figure 1). The inpatient expenditures were the accumulation of all the inpatient costs reported by patients (Figure 1). For the patients without hospitalization, the inpatient expenditures were recorded as zero. All costs were measured in the Chinese Yuan Renminbi (CNY).

The annual direct costs of SLE included medical and non-medical costs. The former included the charges for the physicians’ visits, drugs, laboratory tests and imaging examinations, antibody examinations, physiotherapy, and other services (e.g., purchase of medical aids), hospitalizations, and surgeries. The latter were the costs incurred in the pathway to care and/or to access the services, including transportation, food, and accommodation costs, as well as the informal care costs.

### 2.5. Statistical Analysis

The number of outpatient visits reported by patients provided an extreme value in our study, which might overestimate the total costs. We used values of the 99th percentile to replace the extreme value in order to reduce its influence. The direct medical, non-medical, and total direct costs were described by both mean (standard deviation) and median (interquartile range, IQR), as was each of the cost domains.

Cost data are typically right-skewed distribution because a few patients incurred particularly high costs. The bootstrap method was popular for the skewed data. In this study, cost data are also represented as arithmetic mean (bootstrap 95% CI). The arithmetic means of the costs were calculated using the bootstrap method with 1000 bootstrap samples by resampling with replacement. A 95% confidence interval (CI) for the mean was estimated by the empirical bootstrap method and considered the right-skewed distribution of the costs [26,27]. The comparisons of the direct costs for each subgroup were conducted using the Mann–Whitney U test or Kruskal–Wallis’s test. A *p*-value of <0.05 was considered statistically significant.

The drivers associated with annual direct costs were identified using multivariate regression models. Annual direct costs (dependent variable) were normalized using a logarithmic transformation. The stepwise method was used to identify primary factors related to the direct costs. A *p*-value of 0.05 was used to add a variable to the model or to remove a variable from the model.

All the statistical analyses were conducted using SAS 9.4 software (SAS Institute, Cary, NC, USA).

## 3. Results

### 3.1. Socio-Demographic and Clinical Characteristics

There were 1778 SLE patients in this study, who were from 101 state-owned hospitals belonging to 27 provinces in China (Figure 2). The majority of these patients were females (92.58%), with a mean age of 33.8 ± 10.8 years old, and a median duration of SLE of 4.9 years (IQR: 1.8–9.5 years). A total of 63.8% of them were in an active disease state, with 10% of patients exhibiting moderate or severe diseases, about 77.3% possessed at least two damaged organs, 8.3% used biologics as a treatment and 18.9% were treated without glucocorticoids. The median income of the patients was CNY 27,600 (IQR: CNY 0–50,000) during the last year, while about 630 patients (35%) came from low-income populations (less than CNY 2000/month, according to the data released by the National Bureau of Statistics of China in 2021 [28]) (Table 1).

### 3.2. Direct Cost of SLE Patients

Table 2 illustrates the total per patient–year direct costs. The median annual direct costs of SLE per patient was CNY 14,867 (IQR: CNY 6820–31,180), and the arithmetic mean was CNY 29,727 (bootstrap 95% CI: CNY 27,384 to 32,319), in which more than 85.5% were medical costs (Table 2). Of the direct medical costs, drug expenses accounted for the greatest proportion (53%), followed by the expense for laboratory tests, imaging, or antibody examinations (27%). Of the direct non-medical costs, the proportion of transportation expenses was 41%, followed by food expenses (25%), and accommodation expenses (24%) (Figure 3).

### 3.3. Cost-Driving Factors

As shown in Table 3, the higher direct costs were found in the patients with shorter durations (especially the newly treated patients), hospitalization during last year, ANA positive, active disease state (especially moderate to severe disease states), damaged organs (especially those system involved in peripheral vascular, serositis, renal, cardiovascular, or pulmonary systems), and using biologics or moderate-to-high dose glucocorticoids (all the comparisons, *p* < 0.05, respectively).

The findings in the regression model are represented in Table 4. The primary cost-drivers focused on moderate to severe diseases, use of biologics, treatment with moderate or high dose glucocorticoids, hospitalization, and peripheral vascular, cardiovascular, and/or renal systems involved, all of which substantially increased the annual direct costs. For the patients who were hospitalized last year compared to those non-hospitalized, there was an increase of 239% (amount: CNY 19,150, *p* < 0.01) in annual direct costs controlling for covariates. The direct costs also increased by nearly 50% among those undergoing biologics treatment (41%, amount: CNY 3248, *p* < 0.01), or cardiovascular damage (48%, amount: CNY 3826, *p* < 0.01). Moreover, compared to those without glucocorticoids, the direct costs increased one-fourth when using moderate-dose glucocorticoids and one-third when using high-dose glucocorticoids, however, it remained similar to those using low-dose glucocorticoids. However, the direct costs of SLE decreased by 12% compared with the lack of health insurance (amount: CNY −978, *p* = 0.04).

## 4. Discussion

To the best of our knowledge, this is the largest multi-center, cross-sectional study that has estimated the direct cost and the drivers of costs associated with SLE in China. The average annual direct cost per patient was estimated in this study at CNY 29,727, accounting for nearly 85% of the per capita disposable income (CNY 35,128) of residents in 2021, released by the National Bureau of Statistics of China [28]. The bulk of the direct costs were the medical costs (86% of the total), majorly allocated to the expense of medication and the laboratory or imaging, or antibody tests. The moderate to severe disease activities, use of biologics, hospitalization, treatment of moderate or high dose glucocorticoids, peripheral vascular, cardiovascular, and/or renal systems involved was found to substantially increase the direct costs, while the use of health insurance slightly decreased the direct costs of SLE. All of these results provide the basic evidence needed to further research and describe the profiles of economic burden on society, which will optimize the treatment strategies and achieve the ultimate goal of improving patients’ disease outcomes.

Until now, the evidence of SLE costs originated from several developed countries or regions, including several European countries, the United States, Canada, Japan, Korea, Hong Kong, and Taiwan [8,9,10,11,12,13,14,15,16,29]. However, it was inappropriate to compare these reported costs between counties or regions due to variations in the methods used for data collection, the monetary valuation of the included resources, the healthcare delivery systems involved, and the pricing of the respective health services, as well as the features of the patients [30]. Compared with two previously published studies in China, the average annual direct cost estimated in this study was lower than the CNY 33,899 previously estimated in the study of Shanghai (eastern region) [24] and higher than the CNY 20052.8 in the study conducted in Anhui province (central region) [25]. The patients in our study were recruited from multiple centers, covered in most provinces in China, and the major clinical characteristics of these patients were similar to those of Chinese SLE patients from CSTAR, as reported in 2013 [31]. There was a better representation of SLE patients in this study than those of the studies conducted in the Shanghai and Anhui provinces. The results of our study objectively described the whole profile of costs in current Chinese SLE patients.

The findings in our study indicate that the economic pressures of SLE on the patients were obvious because the average annual direct cost accounted for nearly 85% of the per capita disposable income in China [28]. Only less than 20% of patients in the study reported having been hospitalized during the last year, which also means that outpatient expenditure was the main resource of the direct cost. According to regulations of health insurance in China, the outpatient expenses are still reimbursed, although only in a very low proportion of the health insurance, which slightly varied between provinces. In our study, more than half of the patients reported that all outpatient expenses were paid for by themselves. Even for those who were hospitalized in our study, approximately 50% of the inpatient expenses should be paid by themselves. These results implied most patients might face financial hardship due to SLE with chronic and incurable features, especially for those living close to or below the poverty line, who had families with severe disease forms and/or major organ damage. Unfortunately, several studies also reported that poverty was related to higher disease activity, increased organ damage, and higher mortality in SLE patients of varying ethnic backgrounds [32,33,34]. Moreover, a number of biological therapies are currently being developed and used in SLE treatment. Although less than 10% of patients in our study used biologics, the direct cost of SLE patients using biologics was found to be more expensive than those who were not. It also implied that the expenses of novel therapies (e.g., biologics) might delay or prevent access to specialist biologics and limit the treatment options available to patients. Therefore, it is necessary to evaluate, in the future, whether the potential benefits of these therapies will be commensurate with their costs. It is important to identify the target patients that may be candidates for novel lupus treatments, from the perspective of health economics.

Rarely have studies focused on the cost-drivers of SLE in China. The dominant cost-drivers were found to be hospitalization, use of biologics, moderate to high dose glucocorticoids, major organ involvements (e.g., cardiovascular, kidney, peripheral vascular system), disease status, and severity of the disease. These findings were consistent with previous studies [8,14,20,21,35,36]. The findings provided evidence that the interventions achieved to effectively control disease activity, prevent disease flareups, delay, or slow disease progression might potentially save large amounts of direct costs, especially attributable to disease progression and cumulative organ damage [37,38]. The long disease duration in this study was found to slightly decrease the direct costs of SLE, but no association was identified when controlling for covariates (e.g., disease status and severity, etc.). Lower direct costs related to the longer disease durations were also identified in the results by Huscher et al. [39] and Zhu et al. [15], while the opposite results were observed by Panopalis et al. [40]. Adherence was an important factor in the management of SLE associated with both costs of SLE and the outcome of treatment. It was reported in previous studies that patients were more likely to be persistent in the first and second-year follow-ups, especially for those with low economic status [41]. The study of SLE conducted in Egypt also found that long disease duration was negatively related to medication adherence [42]. Furthermore, poverty was found to drive medication non-adherence [41]. All of these might be partly attributable to why patients with short disease durations were found to possess higher direct costs in our study. Lastly, the difference in annual direct cost between age subgroups was not found in this cross-sectional study, which would be due to the similar clinical management and treatment of SLE patients in China.

This was the first study to provide evidence of the average annual direct costs and the cost-drivers related to SLE. Furthermore, it represents the largest multi-center, representative sample of SLE patients from 101 state-owned hospitals covered in 27 provinces in China. Moreover, bootstrapping analysis was used to estimate confidence intervals of costs with skewed distribution to provide intuitive and accurate standard errors in our study. However, there were still limitations in our study. Firstly, the estimation was based on data collected from patient-self reports, which might introduce bias. In our study, in order to improve the accuracy as much as possible, we only invited the patients whose clinical records of CRDC were updated by the clinicians who they visited during the last 30 days, and participants were requested to recall the expense information of this recent visit in detail alongside the frequency of their visits over the previous year. Moreover, at the initial stage, 53 patients were randomly selected to collect information repeatedly over two weeks to assess the agreement of information. Secondly, it was difficult to distinguish, clearly, the costs that were attributable directly to SLE and which were attributable to comorbidities (which might not be associated with SLE), thus, we incorporated all health resource utilizations and expenses of SLE patients during the last year, regardless of the cause. Further, the average and bootstrap 95% CIs were used to estimate the annual direct cost with the positively skewed distribution. This might cause a risk of overestimation in this study. Thirdly, the variation of inpatients between hospitals was not considered in the data analysis. This was because the sample sizes varied between hospitals, from a few to hundreds of patients, which led to wide variations in the rate of inpatients, especially in the hospital with only a small sample size. All the sites belonged to western, central, and eastern regions of China, and the rates of inpatients among the patients from the western, central, and eastern regions were 22.31%, 20.51%, and 17.81%, respectively, which were not significantly different (*p* = 0.14). Fourthly, about 8.32% of patients used biologics in our study, which is slightly higher than the 5.66% that utilized biologics among the CSTART cohort in 2021. Additionally, this might lead to a slight overestimation of the annual direct cost of SLE in our study. Lastly, the indirect and intangible costs of SLE were not assessed in this study (a further study will describe these in detail), which would lead to an underestimation of the cost of SLE in China.

## 5. Conclusions

In summary, this study provided reliable insight into the annual direct costs and the drivers related to SLE in China. The findings indicated that SLE imposes significant financial pressures on individual patients. The efforts focusing on preventing the occurrence of flareups and limiting disease progression were recommended to further reduce the direct costs of SLE. Further economic evaluations of the novel strategies of therapies and management of SLE are needed to help clinicians and policymakers in their decision-making.

## Figures and Tables

**Figure 1 ijerph-20-03522-f001:**
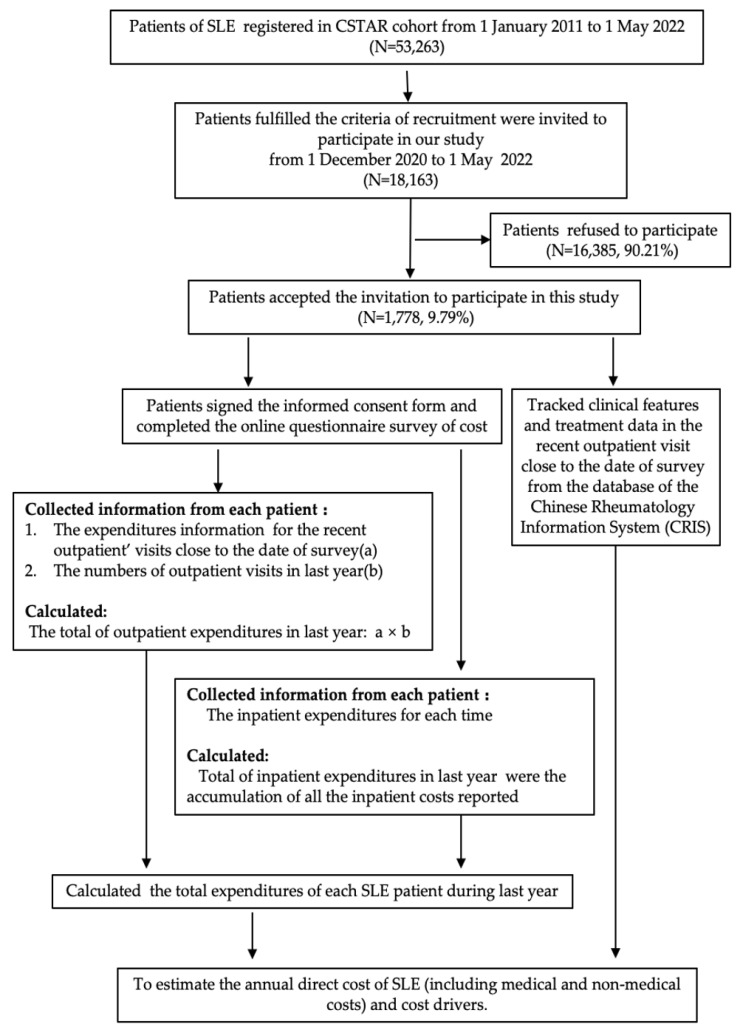
A flow chart of study participants and its design.

**Figure 2 ijerph-20-03522-f002:**
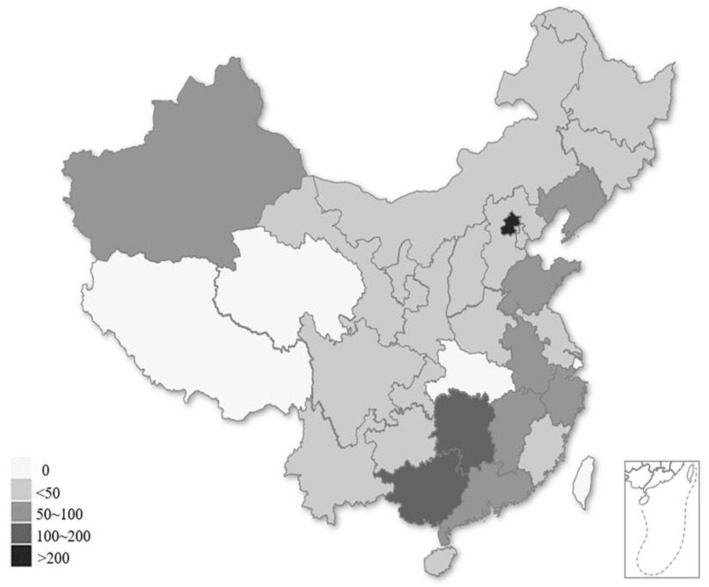
Provincial distribution of the SLE patients in our study, according to the hospitals at which they were registered.

**Figure 3 ijerph-20-03522-f003:**
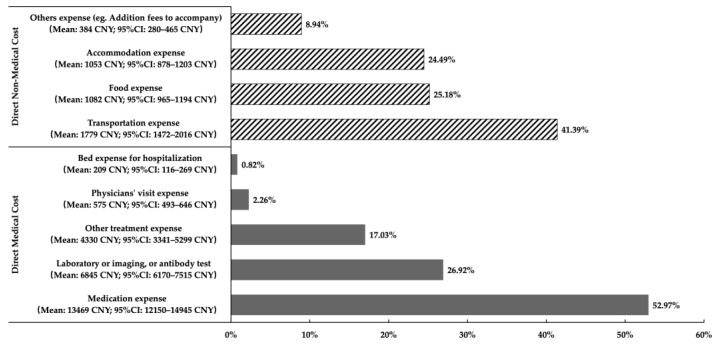
Proportion of direct medical costs and non-medical costs of SLE patients in China, calculated by the arithmetic means.

**Table 1 ijerph-20-03522-t001:** SLE patient demographics and clinical profiles.

	Total (n = 1778)
Age (years), mean (SD)	33.8 (10.8)
Gender, females, n (%)	1646 (92.58%)
Disease duration (years), median (p25–p75)	4.9 (1.8–9.5)
Newly treated patients, n (%)	133 (7.48%)
Regions	
Western	372 (20.92%)
Central	429 (24.13%)
Eastern	977 (54.95%)
Medical insurance, n (%)	1508 (84.81%)
Patients’ incomes during last year (CNY), median (p25–p75)	27,600 (0–50,000)
Numbers of outpatient visits during last year, median (p25–p75)	5 (3–8)
Hospitalization during last year, n (%)	345 (19.40%)
Hospitalization days for inpatients, median (p25–p75)	11 (7–17)
Positive antinuclear antibody (ANA), n (%)	1691 (95.11%)
Positive anti-double-stranded DNA (dsDNA), n (%)	1359 (76.43%)
Abnormal liver function, n (%)	380 (21.37%)
Low complement, n (%)	511 (28.74%)
Physician’s global assessment, median (p25–p75)	0.80 (0.40–1.10)
SLEDAI scores, median (p25–p75)	2.00 (0–4.00)
Severity of the disease, n (%)	
Mild (SLEDAI score < 8)	1600 (89.99%)
Moderate to severe (SLEDAI score ≥ 8)	178 (10.01%)
Disease activity state, n (%)	
Remission or LLDAS	643 (36.16%)
Active disease	1135 (63.84%)
System involvement, n (%)	
Peripheral vascular system	919 (51.69%)
Serositis	740 (41.62%)
Skin system	1154 (64.90%)
Musculoskeletal system	942 (52.98%)
Renal system	685 (38.53%)
Neuropsychiatric system	115 (6.47%)
Gastrointestinal system	38 (2.14%)
Ocular system	22 (1.24%)
Cardiovascular system	127 (7.14%)
Pulmonary system	17 (0.96%)
Number of organs damaged, n (%)	
0 or 1	404 (22.72%)
2 or more	1374 (77.28%)
Treatment of biologics, n (%)	148 (8.32%)
Treatment of glucocorticoids, n (%)	1442 (81.10%)
0 mg/day (none)	336 (18.90%)
≤7.5mg/day	533 (29.98%)
>7.5 and ≤15 mg/day	510 (28.68%)
>15 mg/day	399 (22.44%)

**Table 2 ijerph-20-03522-t002:** Annual direct cost of SLE per patient (Unit: CNY).

	Annual Direct Medical Cost	Annual Direct Non-Medical Cost	Total Annual Direct Cost
Mean ± SD	25,520 ± 49,592	4298 ± 10,938	29,818 ± 53,389
Median (p25–p75)	11,698 (4869–25,644)	1650 (568–4270)	14,867 (6820–31,180)
Arithmetic mean (bootstrap)	25,429	4298	29,727
Bootstrap 95% CI	23,346 to 27,733	3745 to 4762	27,384 to 32,319
Proportion of total direct cost	85.5%	14.5%	100%

**Table 3 ijerph-20-03522-t003:** Details of the annual direct cost of SLE patients in China (Unit: CNY).

**Features**	**N**	**Arithmetic Mean** **(Bootstrap)**	**95% CI** **(Bootstrap)**	**Median (IQR)**	***p*-Value ***
Gender					0.81
Male	132	27,225	21,270–33,300	15,424 (5429–24,578)	
Female	1646	29,928	27,470–32,730	14,833 (6920–30,764)	
Age groups					0.50
<18 years	55	36,347	11,533–53,543	17,720 (7968–33,000)	
18–39 years	1268	29,417	26,473–32,453	14,292 (6899–29,717)	
≥40 years	455	29,785	25,722–33,671	15,275 (6522–35,093)	
Disease duration					<0.001
≤3 years	651	35,064	30,794–39,189	18,120 (8405–38,090)	
3–8 years	555	26,454	21,568–31,057	12,276 (5875–26,264)	
>8 years	572	26,822	23,033–30,352	12,836 (6349–30,263)	
Newly treated patients					<0.001
Yes	133	37,870	29,540–45,038	25,162 (9231–48,000)	
No	1645	29,071	26,578–31,755	14,165 (6650–30,223)	
Region					0.24
Western	372	31,075	26,530–35,979	16,558 (7315–32,999)	
Central	429	29,057	24,463–33,994	14,232 (6772–30,048)	
Eastern	977	29,507	25,908–32,796	13,912 (6724–30,760)	
Health insurance					0.85
No	270	28,369	22,106–33,699	14,158 (7433–30,223)	
Yes	1508	29,972	27,317–32,814	14,970 (6656–31,496)	
Hospitalization during last year					<0.001
Yes	345	64,755	56,272–72,590	40625 (23,272–71,727)	
No	1433	21,289	19,089–23,483	11,436 (5667–22,384)	
ANA					0.01
Positive	1691	30,301	27,838–33,020	15,231 (7000–32,100)	
Negative	87	18,556	12,889–23,761	10,401 (5040–24,200)	
dsDNA					0.53
Positive	1359	29,613	27,178–32,371	14,833 (6950–31,385)	
Negative	419	30,093	23,515–35,909	15,450 (6348–30,280)	
Abnormal liver function					0.04
Yes	380	31,398	26,219–36,323	16,879 (7473–35,901)	
No	1398	29,272	26,268–32,177	14,072 (6566–30,512)	
Low complement					0.07
Yes	511	33,279	28,820–38,023	16,379 (6922–34,707)	
No	1267	28,299	25,529–31,295	14,232 (6800–30,312)	
Severity of the disease					<0.001
Mild	1600	27,498	25,136–30,097	13,755 (6576–29,213)	
Moderate to severe	178	49,720	39,578–59,592	27,269 (11,730–55,497)	
Disease Activity State					<0.001
Remission or LLDAS	643	21,337	18,932–23,841	10,600 (5006–23,220)	
Active disease	1135	34,492	31,095–38,342	17,500 (8216–36,454)	
System involvement					
Skin system					0.67
Normal	624	30,645	25,997–35,103	15,167 (6872–31,724)	
Abnormal	1154	29,231	26,454–32,076	14,549 (6800–31,040)	
Musculoskeletal system					0.88
Normal	836	30,730	26,796–34,904	15,264 (6749–31,949)	
Abnormal	942	28,839	26,029–31,797	14,782 (6920–30,428)	
Peripheral vascular system					<0.001
Normal	859	24,699	22,318–27,395	12,511 (6155–26,460)	
Abnormal	919	34,432	29,980–38,455	17,243 (7394–35,712)	
Serositis					<0.001
Normal	1038	25,254	22,088–28,390	12,501 (6150–25,750)	
Abnormal	740	35,994	32,291–40,338	18,350 (7940–40,312)	
Renal system					<0.001
Normal	1093	25,864	22,770–28,962	13,030 (6095–27,114)	
Abnormal	685	35,879	32,177–40,254	17,990 (7980–37,958)	
Neuropsychiatric system					0.47
Normal	1663	29,014	26,758–31,551	14,743 (6772–31,000)	
Abnormal	115	40,069	25,188–53,849	15,738 (7050–37,958)	
Gastrointestinal system					0.30
Normal	1740	29,344	26,945–31,906	14,833 (6851–30,900)	
Abnormal	38	47,167	21,122–67,857	17,645 (6520–51,660)	
Ocular system					0.46
Normal	1756	29,649	27,211–32,287	14,833 (6801–31,080)	
Abnormal	22	35,918	14,917–53,009	17,754 (8000–40,618)	
Cardiovascular system					<0.001
Normal	1651	29,069	26,664–31,689	14,100 (6559–30,428)	
Abnormal	127	38,321	28,127–46,539	19,935 (11,220–43,701)	
Pulmonary system					0.04
Normal	1761	29,499	27,153–32,159	14,743 (6803–31,120)	
Abnormal	17	53,274	23,230–79,140	26,784 (13,345–98,400)	
Number of Organs damaged					<0.001
0 or 1	404	23,048	17,132–27,992	12,105 (5825–23,546)	
2 or more	1374	31,693	28,941–34,647	15,761 (7196–34,334)	
Biologics					<0.001
Yes	148	48,817	33,271–60,434	26,890 (11,770–55,241)	
No	1630	27,996	25,868–30,462	13,755 (6559–29,500)	
Dose of Glucocorticoids					<0.001
0 mg/day (Unused)	336	23,515	19,851–26,974	11,884 (5483–26,238)	
≤7.5mg/day	533	18,903	16,665–21,050	11,050 (5244–20,589)	
>7.5mg/d and ≤15mg/day	510	34,280	29,092–39,751	16,340 (7941–35,712)	
>15mg/day	399	43,608	35,702–50,329	23,270 (10,953–48,216)	

* Method: Wilcoxon Rank Sum Test.

**Table 4 ijerph-20-03522-t004:** Results of the univariate and multiple linear regression models.

	Univariate Linear Regression Model			Multiple Linear Regress Model (Method: Full)			Multiple Linear Regress Model (Method: Stepwise)			% of Increments or Decrements (Stepwise)	
	**β**	**SE**	** *p-* ** **Value**	**β**	**SE**	** *p* ** **-Value**	**β**	**SE**	** *p* ** **-Value**	**% ***	**Amount (CNY) #**
Intercept	N. A	N. A	N. A	8.49	0.36	<0.01	8.99	0.16	<0.01	—	8022 (ref)
Female (ref = Male)	0.06	0.11	0.56	0.12	0.09	0.21					
Age groups (ref: ≥40 years)											
<18 years	0.10	0.17	0.53	−0.03	0.15	0.82					
18–39 years	−0.04	0.06	0.53	−0.07	0.06	0.23					
Disease duration (ref = ≤3 years)											
3–8 years	−0.36	0.07	<0.01	−0.08	0.07	0.23					
>8 years	−0.31	0.07	<0.01	−0.10	0.07	0.15					
Region (ref: West)											
Central	−0.12	0.08	0.15	−0.09	0.07	0.21					
Eastern	−0.11	0.07	0.13	−0.13	0.07	0.04					
Health insurance (ref: None)	−0.03	0.08	0.67	−0.15	0.07	0.03	−0.13	0.07	0.04	−12.2	−978
Hospitalization during last year (ref: None)	1.32	0.06	<0.01	1.19	0.06	<0.01	1.22	0.06	<0.01	238.7	19,150
Newly treated patients (ref: Treated patients)	0.36	0.11	<0.01	−0.02	0.10	0.84					
ANA positive (ref: Negative)	0.30	0.13	0.02	0.20	0.11	0.08					
dsDNA positive (ref: Negative)	0.04	0.07	0.54	0.04	0.06	0.53					
Abnormal liver function (ref: Normal)	0.13	0.07	0.06	0.05	0.06	0.43					
Low complement (ref: No)	0.11	0.06	0.08	−0.03	0.06	0.63					
Moderate to severe (ref: Mild)	0.60	0.09	<0.01	0.27	0.09	<0.01	0.26	0.08	<0.01	29.7	2382
Active disease State (ref: Remission or LLDAS)	0.46	0.06	<0.01	0.06	0.08	0.47					
System involvement											
Skin system (ref: Normal)	−0.02	0.06	0.71	−0.01	0.06	0.83					
Musculoskeletal system (ref: Normal)	−0.01	0.06	0.87	−0.03	0.05	0.60					
Peripheral vascular system (ref: Normal)	0.22	0.06	<0.01	0.12	0.05	0.02	0.13	0.05	<0.01	13.9	1114
Serositis (ref: Normal)	0.33	0.06	<0.01	0.01	0.10	0.90					
Renal system (ref: Normal)	0.30	0.06	<0.01	0.20	0.10	0.04	0.20	0.05	<0.01	22.1	1776
Neuropsychiatric system (ref: Normal)	0.10	0.11	0.38	0.01	0.10	0.94					
Gastrointestinal system (ref: Normal)	0.26	0.19	0.17	0.13	0.17	0.45					
Ocular system (ref: Normal)	0.22	0.25	0.38	0.10	0.22	0.65					
Cardiovascular system (ref: Normal)	0.43	0.11	<0.01	0.42	0.10	<0.01	0.39	0.09	<0.01	47.7	3826
Pulmonary system (ref: Normal)	0.65	0.29	0.02	0.26	0.25	0.30					
2 or more damaged organs (ref: 0 or 1)	0.25	0.07	<0.01	−0.01	0.08	0.89					
Biologics (ref: Unused)	0.59	0.10	<0.01	0.34	0.09	<0.01	0.34	0.09	<0.01	40.5	3248
Glucocorticoids (ref: Unused)											
≤7.5mg/day	−0.09	0.08	0.24	−0.01	0.07	0.92	−0.02	0.07	0.73	−2.0	−159
>7.5mg/d and ≤15mg/day	0.34	0.08	<0.01	0.19	0.09	0.04	0.23	0.07	<0.01	25.9	2074
>15mg/day	0.60	0.08	<0.01	0.24	0.10	0.01	0.30	0.08	<0.01	35.0	2807

* Calculated as: (exp(coefficient) − 1) × 100; # Calculated as: (Exp(coefficient) − 1) × exp (Intercept); Full model: R^2^ = 0.261; Stepwise model: R^2^ = 0.254.

## Data Availability

The data that support the findings of this study are available from the corresponding author, Wang Y.H. and Zeng X.F. upon reasonable request.

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
