# Peer review of "Annual Direct Cost and Cost-Drivers of Systemic Lupus Erythematosus: A Multi-Center Cross-Sectional Study from CSTAR Registry"

_ijerph, 2023, doi:10.3390/ijerph20043522_

Round 1

Reviewer 1 Report

SLE is an intractable disease that is difficult to cure completely, and the patient burden, including the financial burden, is significant.

This study provides evidence for the proposal of health insurance to reduce the financial burden of SLE patients and their families in China.

In this sense, this study is highly significant.

I think that the contents of this paper are generally persuasive and there are no major problems.

I show the following minor comments.

1.P3L96-99

It was in the number of patients and hospitals in this study, but what percentage of the total in China? That would allow us to confirm the validity of the sample size.

2.P3L106

The term "loss of productivity due to SLE patients and their attendants, etc." is to be understood as the amount of salary for leave and so on. Was it for hospital visits and hospitalization?

3.P4L167-169

 You use a p-value of 0.10 for the stepwise method. Why do you use a p-value of 0.10, when you generally use a p-value of 0.05, including other parts of this study? It seems arbitrary.

Or why not consider using other evaluation indices such as AIC?

Author Response

Dear Reviewer,

Thank you very much for your time involved in reviewing the manuscript and your very encouraging comments. We also appreciate your clear and detailed feedback. In the remainder of this letter, we discuss each of your comments individually along with our corresponding responses. To facilitate this discussion, we first retype your comments in italic font and then present our responses to the comments.

Comment 1(P3L96-99): It was in the number of patients and hospitals in this study, but what percentage of the total in China? That would allow us to confirm the validity of the sample size.

Response:

Thank you for your comments. During the period of our study, 18163 patients, who were fulfilled the 1997 SLE classification criteria or 2012 SLICC classification criteria for SLE and updated the clinical records on CRDC during last 30 days, were invited to participate in this online survey. Finally, 1778 patients (9.79% of 18163 patients) accepted the invitation and completed the survey. We updated as “During the period of study, there were the 18163 patients who met the criteria of recruitment. Of them, a total of 1778 patients (about 9.79%), from 101 state-owned hospitals in 27 provinces of China, accepted the invitation and participated in this survey (Figure 1).”(on Page 3 line 97-101.) in the revised manuscript. Moreover, in order to show more clearly, we also added the new flowchart of study participants and design to the revised manuscript (Figure 1).

Comment 2 (P3L106): The term "loss of productivity due to SLE patients and their attendants, etc." is to be understood as the amount of salary for leave and so on. Was it for hospital visits and hospitalization?

Response:

Thank you for your kind comment. “loss of productivity” was not appropriate expression in original manuscript. It was revised as “work loss days of absence related to medical visits or inpatient” in the revised manuscript (Page 3 Line 112-113).

Comment 3(P4L167-169): You use a p-value of 0.10 for the stepwise method. Why do you use a p-value of 0.10, when you generally use a p-value of 0.05, including other parts of this study? It seems arbitrary. Or why not consider using other evaluation indices such as AIC?

Response:

Thank you for your good suggestion. In the revised manuscript, we used the p value of 0.05 (Page 4, Line 187) as the criteria in stepwise method in order to be consistent with other parts of this study, and all the results in Tables 4 were unchanged.

Thank you for all your time involved and this great opportunity for us to improve the manuscript. we appreciate for your comments and valuable suggestions to improve the quality of our manuscript. We hope you will find this revised version satisfactory.

Best regards.

Yanhong Wang

Reviewer 2 Report

While the authors have collected extensive data for their research project, and published some very interesting facts, there are some concerns I would like to raise?

1. The authors have clearly stated that the aim of the study was to estimate the direct costs with subsequent monetary burden.  From information presented, 85% of participants utilized medical insurance, the other 15% are they attending state-owned hospitals?  Is this proportion of private medical care an accurate reflection of medical care in China?

If 85% of study population were medical insurance members, did this study exclude participants from low-socioeconomic status?  One would predict that in the latter group, the co-morbidities would be increased and more severe.   

2.  According to data published, 50% of participants had to pay for inpatient therapy as not all treatments were not covered by the medical insurance.  Will the rates of inpatient therapy not differ in the 101 sites from which participants were recruited?  Will this not impact the monetary values estimated for cost related to SLE?

3.  Economic burden is mentioned in the article, was there no indication to include the measurement of QALY and DALY?  This would have facilitated your public health intervention recommendations, you state in the discussion.

4.  A very small proportion, <10% were on biologics.  Is this common practice in China?  The low utilization of biologics could be significant when active (1135) vs non active (643) disease was calculated.  As this had a direct impact on the cost of illness?

Author Response

Dear Reviewer,

Thank you very much for your time involved in reviewing the manuscript and your very encouraging comments. We also appreciate your clear and detailed feedback. In the remainder of this letter, we discuss each of your comments individually along with our corresponding responses. To facilitate this discussion, we first retype your comments in italic font and then present our responses to the comments.

Comment 1: The authors have clearly stated that the aim of the study was to estimate the direct costs with subsequent monetary burden. From information presented, 85% of participants utilized medical insurance, the other 15% are they attending state-owned hospitals?  Is this proportion of private medical care an accurate reflection of medical care in China? If 85% of study population were medical insurance members, did this study exclude participants from low-socioeconomic status?  One would predict that in the latter group, the co-morbidities would be increased and more severe.  

Response:

Thank you for your comments. All the 101 hospitals in our study were state-owned hospitals. Due to the lack of patients from private hospitals, it was difficult to assess its impacts on our results. Also, we completely agree that low-income patients might be accompanied by more comorbidities and severe disease activity. In our study, about 630 patients (35%) came from low-income populations (less than 2,000 CNY/month, ac-cording to the data released by National Bureau of Statistics of China in 2021). It was added to the results in the revised manuscript (Page 4, Line 199-201).

Comment 2: According to data published, 50% of participants had to pay for inpatient therapy as not all treatments were not covered by the medical insurance.  Will the rates of inpatient therapy not differ in the 101 sites from which participants were recruited?  Will this not impact the monetary values estimated for cost related to SLE?

Response:

Thank you for your comments. We completely agreed with you that inpatient therapy was the important factor associated with the cost of SLE. In fact, we also found in our results that the increment of annual direct cost was 19,150 CNY for those with inpatient therapy compared to those with only outpatient therapy when controlling for other factors. However, the variation of inpatient between hospitals were not considered in data analysis. It was because that the sample sizes varied in hospitals from a few to hundreds of patients, which leaded to the widespread variations of the rate of inpatient, especially in the hospital with small sample size. All the sites were belonged to Western, Central, and Eastern region of China, and the rates of inpatient among the patients from Western, Central, and Eastern region, were 22.31%, 20.51% and 17.81%, that was no significant difference (p=0.14). So, we didn’t further control for the variation from the level of sites or regions in our analysis. It was added to the limitations in the revised manuscript. (on page 13 line 359-364).

Comment 3: Economic burden is mentioned in the article, was there no indication to include the measurement of QALY and DALY?  This would have facilitated your public health intervention recommendations, you state in the discussion.

Response:

Thank you for your kind comments. “economic burden” was not appropriate expression in the original manuscript. The aim of this study was to estimate the annual direct cost of per SLE patient and cost-drivers in China. The assessment of economic burden of SLE need more information (such as prevalence rates, mortality rate, et al) on the population level, except for annual direct cost per patient. It will be the issue of concern in our further studies. So, we revised this expression in the revised manuscript. Thank you very much.

Comment 4: A very small proportion, <10% were on biologics.  Is this common practice in China?  The low utilization of biologics could be significant when active (1135) vs non active (643) disease was calculated. As this had a direct impact on the cost of illness?

Response:

Thank you for your comments. Biologics was considered to be used for the patients who were refractory, intolerant to glucocorticoids and/or immunosuppressive therapy or relapsed in China. Also, only Belimumab was approved by the China Food and Drug Administration (CFDA) for the treatment of SLE currently. So, low utilization might be common among SLE patients in China. About 8.32% patients used biologics in our study, which slightly higher than 5.66% of biologics utilization among the CSTART cohort in 2021, that might slightly overestimate the annual direct cost of SLE in our study. We added it to the limitation in the discussion (Page 13 Line 365-367).

As you were concerned, the disease activity state was associated with the utilization of biologics, and both of them increased the annual direct cost of SLE, which were also consistent with our findings (Table 4). There was also significant difference in utilization of biologics between active (123/1135, 10.81%) and non-active disease activities (245/643, 3.89%) in our study. Furthermore, among the 1135 patients with active disease activities, the patients with server active state had more higher utilization of biologics (92/957, 17.42%) than those with moderate active state (31/178, 9.61%).

Thank you for all your time involved and this great opportunity for us to improve the manuscript. we appreciate for your comments and valuable suggestions to improve the quality of our manuscript. We hope you will find this revised version satisfactory.

Best regards.

Yanhong Wang

Reviewer 3 Report

Thank you for your work.

I have few comments you might consider in making this work better:

1. The English language is good.

2. Figures should be of higher quality, please use a different approach in copying your figures and results charts.

3. The Materials and Methods section should be improved; I expect to see a conceptualization of the techniques and roadmap of the methodology followed to get the results.

The rest of the paper is just fine after minor spellcheck.

Good Luck

Author Response

Dear Reviewer,

Thank you very much for your time involved in reviewing the manuscript and your very encouraging comments. We also appreciate your clear and detailed feedback. In the remainder of this letter, we discuss each of your comments individually along with our corresponding responses. To facilitate this discussion, we first retype your comments in italic font and then present our responses to the comments.

Comment 1: The English language is good.

Response: Thank you for your comments.

Comment2: Figures should be of higher quality, please use a different approach in copying your figures and results charts.

Response:

Thank you for your comments. We had updated original Figures (Shown as Figure 2 and Figure 3 in revised manuscript) and tried to make them more clearly.

Comment3: The Materials and Methods section should be improved; I expect to see a conceptualization of the techniques and roadmap of the methodology followed to get the results.

Response:

Thank you for your good suggestion. We have tried to revised the methods and hope it could be better. According to your suggestion, the flowchart of study participants and design were added to the revised manuscript (Figure 1). 

Thank you for all your time involved and this great opportunity for us to improve the manuscript. we appreciate for your comments and valuable suggestions to improve the quality of our manuscript. We hope you will find this revised version satisfactory.

Best regards.

Yanhong Wang